# Estimation of the Quality of the Diet of Mexican University Students Using DQI-I

**DOI:** 10.3390/healthcare11010138

**Published:** 2023-01-01

**Authors:** Diana Espino-Rosales, Alejandro Lopez-Moro, Leticia Heras-González, Maria Jose Jimenez-Casquet, Fatima Olea-Serrano, Miguel Mariscal-Arcas

**Affiliations:** 1Department of Nutrition and Food Science, University of Granada, 18071 Granada, Spain; 2Faculty of Physical Education and Sport Sciences, Autonomous University of Chihuahua, Chihuahua 31009, Mexico

**Keywords:** diet, Diet Quality Index-International (DQI-I), girls, Mexican university

## Abstract

The quality of diet can be measured using diet quality indices, based on knowledge of associations between diet and health. The objective of this work was to evaluate whether the International Diet Quality Index is suitable for use as a diet quality index in populations of Mexican university girls. A cross-sectional nutritional survey was conducted at the University of Chihuahua (Mexico), collecting semi-quantitative nutritional information and socio-economic and lifestyle data from a representative sample of 400 women. Mean (Standard Deviation (SD)) age was 21.43 years (SD: 3.72); 59.1% were normal weight, 26.6% overweight, 15.3% obesity. The Diet Quality Index-International (DQI-I) was developed according to the method of Kim et al. (2003) and focused on major aspects of a high-quality diet (variety, adequacy, moderation and overall balance). The total score of Diet Quality Index-International reached 53.86% (SD: 11.43), indicating that the general diet of Mexican women a poor-quality diet. Adequacy scored highest, followed by moderation and variety. Overall balance scored the lowest. Variety: 26.3 % consumed less than 4 food groups daily, only 12.8% take more than 1 serving from each food group, and 50.6% consumed only one source of protein daily. Regarding adequacy, a large proportion of the population reported an intake of proteins, vitamin C, calcium, iron, and fruit greater than 50% of recommendation; the vegetables, fiber and grain groups were less 50%. Poor scores were obtained for total fat and SFA consumption (moderation). No statistically significant differences are observed for any of the variables under study and score of the Diet Quality Index-International: body mass index, weight, physical activity level, education level of father and mother, location of lunch, breakfast considered important, knowledge of nutrition, which allows us to consider a relatively uniform population in its eating habits. These people are close to a Westernized diet, and an intervention in nutritional education would be advisable to improve the intake of unprocessed foods, consume a greater variety of protein sources and significantly reduce consumption of sugary foods and soft drinks. Due to different methodological and cultural factors, the proposed Diet Quality Index-International dietary assessment method does not seem to be useful in the assessment of diet quality in the Mexican university population, so further research is needed to develop a diet quality index adapted to the Mexican population.

## 1. Introduction

Nutritional needs have been the subject of ongoing nutritional studies in the population. An excess of energy and macronutrients in the diet can cause marginal deficiencies in other nutrients. The quality of the diet can be measured using scales called diet quality indices, obtaining a score in a simple way, which compares the quality of one diet to global dietary patterns based on prior knowledge of the associations between diet and health. These instruments quantify the intake of food groups, foods, and nutrients; they also assess factors related to lifestyles and/or determine levels of markers in biological samples with the aim of associating these components with the risk of developing chronic diseases and nutritional deficiencies [1,2].

Various indices have been proposed to assess the quality of the diet of a previously defined population group. Since Kant [3] published the Dietary Diversity Score (range 0–5) based on the number of the main food groups (dairy. meat. cereals. fruits and vegetables) consumed daily. Several indices and modifications have been developed. Drewnowski [4] designed a modified index of the Dietary Diversity Score that considered 164 foods over a period of over 15 days, calling it the Dietary Variety Score [5,6,7]. To assess the quality of a population’s diet, the lifestyle and cultural characteristics of the country must be considered [8] to understand both the nutritional requirements and the foods consumed in the population under study [9,10,11,12,13]. These indices generally allow the evaluation of a population’s eating pattern in relation to whether it adheres to a greater or lesser extent of the recommendations of the dietary guidelines following, as references, widely studied models, as is the case of the Western diet [14,15,16,17].

The evaluation of the diet of the young Mexican population with the use of DQI could guide their nutritional habits. Mexico is experiencing a nutritional transition characterized by a decrease in the prevalence of different forms of malnutrition, and a high prevalence of obesity. In 2022, about 74% of the Mexican population was overweight. Mexico has one of the highest rates of obesity in the OECD. In addition, 34% of the obese population are morbidly obese, which is the highest level of obesity. Diets high in energy, saturated fat, sodium, refined carbohydrates or added sugars, but low in fruits, vegetables, or whole grains, are the main causes of obesity and related diseases. Generally, dietary intake and obesity are marked by socioeconomic situations. For example, in Mexico, the intake of fruits, saturated fats, added fats, and sugars are higher among subjects with high socioeconomic status compared to those with low socioeconomic status The prevalence of obesity was higher in adults with higher socioeconomic status [18,19,20,21].

The aim of the present study was to evaluate the dietary quality of a specific population (university students in northern Mexico, Chihuahua State) using the Diet Quality Index-International (DQI-I) and to relate socioeconomic factors, habits, and body composition to the scores obtained in the index.

## 2. Materials and Methods

### 2.1. Subjects

Out of an initial study population of 420 women, 20 (5%) were excluded for failure to complete the questionnaires, leaving the final sample to include 400 individuals, all who were university students from the School of Social Work (University of Chihuahua. Mex). All of these volunteers signed their informed consent to participation in the study, which was approved by the Science Ethics Committee of the State of Chihuahua (Mexico).

### 2.2. Methods

The questionnaire was completed through face to face individual interviews in a room at the University of Chihuahua. The questionnaire was applied during an academic year (February to May), prior to the COVID-19 pandemic. Each participant was administered with four questionnaires by a specifically trained interviewer (DER) [15,16,17,18,19,20,21,22]. The first questionnaire collected information on age, schooling, family characteristics, and habits, among others. The second was [23] semi-quantitative Food Frequency Questionnaire (FFQ) covering the previous 12 months. This questionnaire recorded the consumption of foods, the number of times per day, week, or month, and the amount consumed each time in g, mL, or domestic measures (e.g., platefuls glassfuls. tea/table spoonful., etc.). Daily food and nutrient intake were estimated (in g or mL) by multiplying the standard serving size of items by the consumption frequency classified as: never = 0; 1–3 times/month = 0.07; 1–2 times/week = 0.21; 3–4 times/week = 0.50; 5–6 times/week = 0.80; 1 time/day = 1; and 2–3 times/day = 2.50. The FFQ designed for the study includes 120 foods classified by food groups (dairy, cereals, eggs, pulses, meat, fish, fats, vegetables, fruits, drinks/infusions, nuts, and others). The third questionnaire was completed in three-day 24 h periods for three non-consecutive days, including one non-working day. It was administered once a month between February and May. At the end, similar information was obtained for three consecutive months. Finally, a questionnaire was used to gather data on anthropometrics and physical activity. Weight (kg) was measured with a floor scale (model SECA 872, Hamburg. Germany), and subjects were barefoot and in light clothes with their height measuredwith a stadiometer (model SECA 214; 20–207 cm), following the CDC Anthropometry Procedures Manual (https://www.cdc.gov/nchs/data/pdf; accessed on: 10 October 2022), classifying participants as normal weight, overweight or obese according to their body mass index (BMI). The Mexican Nutrikal food nutrient database was used to estimate the intake of nutrients and energy, based on the food intake gathered from the semi-quantitative FFQ and calculating the quantity of each nutrient per 100 g of food [24,25,26].

### 2.3. Socio-Demographic Variables

#### 2.3.1. Qualitative Variables

Variables include: weight; physical activity level; education level of father; educational level of mother; location lunch; how important breakfast was considered to be; and knowledge of nutrition. 

#### 2.3.2. Quantitative Variables

Age, weight, and height were measured. BMI was calculated through weight and height; the degree of obesity was based on the WHO classification [1].

### 2.4. Construction of Diet Quality Index-International Briefly

The DQI-I [6] focuses on four aspects of a high-quality diet (variety, adequacy, Moderation, and overall balance). Specific diet components are assessed under each category. These categories help users to identify aspects of their diet that may need improvement. The score for each category is the sum of the scores for each component in that category. The total DQI-I score (range 0–100 points) is the sum of the scores for the four categories.

#### 2.4.1. Variety

Variety was evaluated both as overall variety and as variety of protein sources. The maximum overall variety score was achieved by intake of at least one serving per day from each of the five food groups (meat/poultry/fish/egg, dairy/beans, grains, fruit, and vegetables). The score for the variety of protein sources (meat, poultry, fish, dairy, beans, and eggs) was based on intakes of more than half the serving size per day using data gathered by the FFQ. Portions were based on portion–weight tables for each food group and household measures [25,27].

#### 2.4.2. Adequacy

This category evaluates the adequacy of intake of those dietary elements that are required to protect against under-nutrition and deficiency disorders. The adequacy of fruit, vegetables, grain, and fiber intake is dependent on the energy intake. Thus, for energy intakes of 7118 kJ (1700 kcal), 9211 kJ (2200 kcal), or 11,304 kJ (2700 kcal), the highest score is awarded to the diet that contains 2/3/4 servings of fruit and 3/4/5 of vegetables, respectively. Regarding proteins, intakes were considered adequate when the total energy from protein was >10%. The highest score for grains and fiber was awarded for those intakes/day of >6/9/11 servings of cereals and >20/25/30 g of fiber for the three levels of energy intake, respectively. The highest adequacy scores for iron, calcium, and Vitamin C were derived from the recommendations for the young adult population of Mexicans.

#### 2.4.3. Moderation

Assess the intake of food and nutrients related to chronic diseases, which may need restriction. Total fat intake is assessed with stricter cut-off values than other quality indices. Cholesterol and sodium intake is also assessed. Table sugar, alcohol, oil, etc. are considered “empty calorie foods”.

#### 2.4.4. Overall Balance

The overall balance evaluates the general balance of the diet considering the proportions of energy sources as well as the composition of fatty acids.

#### 2.4.5. Statistical Analysis

The statistical analysis was performed using the SPSS version 25.0 software package (SPSS Inc., Chicago, IL, USA). Means and standard deviations were obtained for each component of the DQI-I. Population percentage values, Student’s *t*-test, and one-way ANOVA were used for associations. Finally, multiple stepwise regressions were also performed (significance of *p* = 0.05).

## 3. Results

### Characteristics of the Study Population

The mean (SD) age of the 400 participants was 21.43 years (SD: 3.72) (range: 18–37 years); 59.1% were normal weight, 26.6% overweight, and 15.3% were classified with type 1 obesity according to the classification of Garrow & Webster [28]. Furthermore, 82.9% do not perform physical activity or do so sporadically, and 16.5% perform physical activity approximately twice a week or more. The studies of the parents are between 35.7% of primary studies, 21.1% of university studies for fathers, 16.9% of primary studies for mothers, and 21.4% of university education.

Table 1 shows the mean intakes calculated from the FFQ results and compares them with international recommendations [2]. The FFQ considered 13 food groups, represented by 120 food items habitually consumed by the general population in Mexico [25]. The nutrients magnesium, iodine, potassium, and folate present average percentages lower than two thirds of the DRI, which represents a nutritional deficit for the population studied.

The FFQ, stepwise multiple regression analysis was performed to study the intake of each nutrient in relation to the total intake, allowing identification of the most important foods in the diet of the population. Table 2 displays data on the following food groups: cereals and grain-based products; roots and tubers with starch; dry grain legumes; nuts and seeds; vegetables; fruit; sugars, syrups, and sweets; meat and poultry; eggs; fish and shellfish; milk and dairy products; oils and fats; and drinks. The 13 food groups identified included 120 food items, and the FFQ allowed estimation of the intake of 20 nutrients, including energy.

The study sample comprised 400 girls’ students. The mean total DQI-I score was approximately 53.86% (SD: 11.43) of the possible score (100%). Adequacy obtained the highest score, then moderation and variety, and finally, the general balance that obtained the lowest score. For the variety score, 26.3% daily consumed less than 4 food groups, and only 12.8% take more of 1 serving from each food group/day and 50.6% daily consumed only one source of protein daily (Table 3). Much of the population reported an intake of proteins, vitamin C, calcium, iron, and fruit greater than 50% of recommendation. The vegetables, fiber, and grain groups were less 50% than recommendations (Table 3). The 62.25% score range of the DQI-I is adequate. The ranges do not reach 50% of the recommendation for the consumption of the fruit group (44.9%) or for the vegetables group; the grain group is at the limit of 50% of the recommendation as it is not met by 45% of the population. On the contrary, the recommendation for protein adequacy (85.6%), Fe (89%), Ca (48.8%), and Vitamin C (76.5%) are above 50% of what is recommended.

The population studied obtained results within the limits for fat and saturated fat in the moderation category. Of the population, 61.6% ingested 300 mg/day of cholesterol and 76.2% of the population complied with sodium intake. Furthermore, 31.4% of the population consumed “empty calories food”. Only 4.9% of the population followed the adequate intake of energy from macronutrients, and 28.0% of the subjects have an adequate fatty acid ratio.

No statistically significant differences were observed for any of the studied variables, BMI (*p* = 0.571), accordance to weight (*p* = 0.781), physical activity level (*p* = 0.521), education level of father (*p* = 0.555), educational level of mother (*p* = 0.411), location lunch (*p* = 0.154), consideration of breakfast as important, (*p* = 0.711), and knowledge in nutrition (*p* = 0.305), which allows us to consider a relatively uniform population in its life habits. There is no statistically significant difference after the ANOVA test applied to the grouping in tertiles of the DQI-I and various nutritional variables and social factors (Table 4). Higher intake of vegetables, fruit, fiber, and nutrients such as riboflavin, fat-soluble vitamins (Vit. A, Vit. D, Vit. E), and lower percent of energy intake from fats and SFA are associated with greater follow-up of the DOI-I; however, the upper tertile is negatively associated with the intake of nutrients of interest in the daily diet such as Ca, Se, and folate. The academic training of the parents (% of university students) does not seem to positively influence the follow-up of the diet quality index, however, mothers with a medium level of education are the ones who mostly follow DQI-I (Table 5).

## 4. Discussion

DQI-I was used in a population of university students from the School of Social Work (University of Chihuahua. Mex) and estimated nutrients for the population under study, compared to the Mexican RDA [25], showed an imbalance for obtaining energy from macronutrients, with a very high participation of lipids in the total energy to the detriment of carbohydrates that are below the recommendation. Equally lower than 2/3 of the RDA are Mg, I_2_, K, and folate.

The foods that contribute the most to the total energy/day are sweets and cereals, followed by meat, soft drinks, and dairy products. This situation coincides with other studies carried out on the young Mexican population [16,29]. Of the population, 50.6% use a single protein source, this makes the diet poor from this point of view; human get proteins in their diet from meat, dairy products, nuts, grains, and beans. The characteristic of various protein sources in the DQI-I as a good, varied diet may be questionable depending on the culture of the population studied. Of the population studied, 26.4% includes fewer than four food groups in their diet, which makes the variety of the diet very poor. These habits coincide with young populations that follow the Western diet model [30,31].

The diet of the studied population was assigned a high adequacy score for protein, iron, Vit. C, and calcium, but low for fruits, vegetables, grains, and fiber that does not meet healthy recommendations [32,33].

According to the scores, the diet is very unbalanced. The DQI-I establishes standards in line with the North American recommendations [34]. Very few subjects meet the DQI-I criteria for energy from fat (<30%). Low scores were also obtained from global balance. More research is needed to establish suitable adaptations of the DQI-I to different populations. Additionally, a poor overall balance follow-up only reaches 25% of the recommended range due to the imbalance in the follow-up of the recommended proportion for macronutrients and for the healthy correlations between MUFA, PUFA, and SFA.

These people are close to consuming a Westernized diet, which is characterized by a high content of proteins (derived from fatty domesticated and processed meats), saturated fats, refined grains, sugar, alcohol, salt, and corn-derived fructose syrup, with an associated reduced consumption of fruits and vegetables. The typical Western (American) diet is low in fruits and vegetables, and high in fat and sodium. Moreover, this diet consists of large portions, high calories, and excess sugar. This excess sugar accounts for more than 13% of the daily caloric intake with beverages constituting 47% of these added sugars [32].

The mean score of the study population was 53.86% of the full score, and lower than the mean DQI-I scores reported in different studies cited by Dalwood [35]. According to the criteria of Kim [6], scores below 60% indicate a poor-quality diet. The highest scores in the present group were for adequacy and moderation. In this population, socio-demographic factors do not sufficiently influence the monitoring of the quality of the diet recommended by DQI-I. No statistically significant differences are observed with BMI; 58% of the population is satisfied with their weight and does not see it influenced by following a quality diet. Breakfast is important for 45% of the population and 62.5% state that they have an average knowledge of nutrition, but as is the case with other population groups [31], this does not seem to influence healthy monitoring of diet quality. The application of knowledge of social relationships on the youth population could broaden the scope of nutritional interventions to promote health in the physical and psychosocial dimensions. [36].

The study of the female university population may present a bias compared to the general population; in the first place, the age range is restricted 21.43 years (SD: 3.72). In addition, being university students at the School of Social Work, it is possible to assume greater knowledge about nutrition and higher cultural level than other groups of similar age and different cultural level, as stated by various studies [37,38]. However, according to the statement of university students, knowledge of nutrition is on average 62.5% of the population and a DQI-I follow-up value of 53.71.

The results of this study suggest that by assessing four major qualities of the diet, the index may also provide useful information for nutrition intervention and education programs in determining which areas of diets require improvement. Because the DQI-I assessed various aspects of diet with a strict set of standards, especially for the fat components, the mean of the DQI-I score in both countries reached only 60% of the highest possible score.

An investigation into the major categories, however, revealed interesting differences between the countries, reflecting each country’s nutritional status and concerns. Diets are multidimensional, and there exist quite different strengths and weaknesses of the dietary patterns of each country, which were well conceptualized by the specific major categories of the DQI-I.

The highest variety score and lowest moderation score in developed countries corresponds to what is observed through the stages of the nutritional transition. Economic development allows for greater food availability, as well as greater food security, therefore allowing greater variety and adequacy of diet [39].

DQI-I gives points for eating meat, poultry, egg yolks, and fatty dairy products. It is widely known that eating an excessive amount of meat is not healthy [40]. WHO has classified red meat as potentially carcinogenic, and all other processed meat as carcinogenic as well [40]. Many studies [41,42] have also shown that eating meat (and also other animal products) increases mortality. According to this knowledge of associations between diet and health, it cannot be stated that DQI-I is a good way measure the quality of diet.

## 5. Conclusions

After the study of nutritional habits, it can be concluded that the diet of the subjects studied is deficient in the consumption of vegetable foods (vegetables, fruits, and grains).

The energy is obtained mainly from carbohydrates and simple sugars (sweets and soft drinks), otherwise known as empty calories. However, the intake of Na, cholesterol, proteins, and Vitamin C are adequate. There are no statistically significant differences between the socioeconomic parameters considered in the study and the DQI-I values.

In general, the diet of this population is monotonous from the point of view of the variety parameter, and it only reaches 44.1% of the parameter defined in DQI-I, and for overalls balance, it only meets 25% of the estimated parameter.

An intervention in nutritional education would be advisable, to improve the intake of plant foods, include a greater variety in protein sources (for example, introduce into your diet combinations of grains and legumes, grains and grains seeds and nuts, legumes seeds and nuts), and significantly reduce the consumption of sugary foods and soft drinks. According to this study, it is necessary to extend the results to male university students and to other population ranges to achieve a better perspective regarding the quality of the diet of the Mexican population.

## Figures and Tables

**Table 1 healthcare-11-00138-t001:** Percentage of daily recommended intake (DRI) of nutrients according to semi-quantitative FFQ result, Herforth et al. (2019) [2].

	Minimum	Maximum	Mean	SD
Energy (Kcal)	1138.72	3131.27	2170.80	425.63
Energy proteins (%)	8.54	23.19	14.45	2.71
Energy carbohydrate (%)	31.91	73.81	48.50	7.43
Energy lipids (%)	18.13	51.94	35.52	6.38
Fiber %RDA	2.29	214.29	45.79	33.00
% RDA				
Calcium (%)	18.76	115.33	74.02	27.40
Iron (%)	20.02	123.23	67.11	23.51
Magnesium (%)	26.54	139.04	59.63	20.37
Iodine (%)	2.16	77.95	27.25	1.74
Selenium (%)	28.20	142.61	105.69	40.24
Sodium (%)	29.01	158.83	96.79	32.76
Potasium (%)	13.63	75.39	37.18	11.91
Phosphorus (%)	33.99	237.14	109.64	39.54
Zinc (%)	35.78	145.45	87.80	36.03
Niacin (%)	21.39	215.98	101.16	49.84
Thiamine (%)	9.14	209.71	72.29	38.08
Riboflavin (%)	22.34	215.26	111.88	52.47
Pyridoxine (%)	16.36	157.68	80.20	33.14
Folate (%)	13.80	107.52	50.45	20.12
Vitamin C (%)	14.75	446.82	135.52	81.86
Vitamin A (%)	52.90	198.78	83.15	47.78
Vitamin E (%)	60.35	94.83	79.16	3.54
Vitamin D (%)	52.56	95.80	78.13	6.35

**Table 2 healthcare-11-00138-t002:** Stepwise multiple r egression results for selected nutrients. showing the foods included in the semi-quantitative Food Frequency Questionnaire.

	Cumulative R^2^	Foods	Cumulative R^2^	Foods	Cumulative R^2^	Foods	Cumulative R^2^	Foods	Cumulative R^2^
Energy Kcal		Protein		Lipids		Carbohydrate		Ca	
Sweets	0.640	Meat	0.772	Fat/Oil	0.679	Cereals	0.646	Dairy product	0.882
Cereals	0.817	Dairy product	0.899	Meat	0.862	Soft drinks	0.869	Sweets	0.943
Meat	0.893	Cereals	0.940	Dairy product	0.941	Sweets	0.943	Legume	0.972
Soft drinks	0.941	Legumes	0.964	Sweets	0.977	Fruit	0.984		
Dairy product	0.967	Fish	0.983						
Mg		Fe		Se		Na		K	
Legumes	0.632	Legumes	0.663	Cereals	0.883	Cereals	0.689	Legumes	0.676
Cereals	0.861	Cereals	0.780	Meat	0.975	Sweets	0.849	Dairy product	0.917
Dairy product	0.948	Meat	0.945	Sweets	0.985	Dairy product	0.952	Cereals	0.989
Meat	0.987								
P		I_2_		Thiamine		Riboflavin		Pyridoxine	
Dairy product	0.705	Dairy products	0.935	Cereals	0.550	Dairy product	0.655	Cereals	0.577
Cereals	0.815	Cereals	0.984	Meat	0.818	Cereals	0.917	Fruit	0.844
Sweets	0.854			Legumes	0.970	Fruit	0.973	Dairy product	0.956
Eggs	0.870					Meat	0.988		
Folate		Niacin		Vit. C		Vit. A		Vit. E	
Legumes	0.886	Cereals	0.954	Fruit	0.766	Dairy product	0.631	Fat/oil	0.940
Cereals	0.981	Dairy product	0.991	Cereals	0.935	Fat/oil	0.792	Legumes	0.060
Dairy product	0.993			Vegetables	0.989	Eggs	0.878	Dairy product	0.970
						Cereals	0.947		

*p* < 0.001 for the cumulative R^2^ value of each nutrient.

**Table 3 healthcare-11-00138-t003:** DQI-I and components [6] in component subcategories (%).

Component	Full Score	Mean	SD	Criteria		%
DQI-I. total	0–100	53.86	11.43			
Variety	0–20	8.82	3.26			
Overall food group variety	0–15	6.60	4.49	≥1 serving from each food group/day	15	12.80
				Any 1 food group missing/day	12	16.60
				Any 2 food groups missing/day	9	22.90
				Any 3 food groups missing/day	6	20.50
				≥4 food groups missing/day	3	26.40
Within-group variety from protein source	0–5	2.22	1.96	≥3 different sources/day	5	26.30
				2 different sources/day	3	23.10
				From 1 source/day	1	50.60
				None	0	0.00
Adequacy	0–40	24.94	4.88			
Vegetable group	0–5	2.03	1.47	≥100% recommendations	5	0.00
				<100–50% recommendations	3	17.00
				<50% recommendations	1	19.90
				0% recommendations	0	63.10
Fruit group	0–5	2.83	2.11	≥100% recommendations	5	38.40
				<50–100% recommendations	3	16.80
				<50% recommendations	1	44.90
				0% recommendations	0	0.70
Grain group	0–5	1.99	1.72	≥100% recommendations	5	15.50
				<100–50% recommendations	3	30.00
				<50% recommendations	1	54.50
				0% recommendations	0	0.00
Fiber	0–5	2.14	1.48	>100% recommendations	5	14.60
				<100–50% recommendations	3	27.40
				<50% recommendations	1	54.90
				0% recommendations	0	0.00
Protein	0–5	4.79	0.67	>100% recommendations	5	89.60
				<100–50% recommendations	3	9.80
				<50% recommendations	1	0.60
				0% recommendations	0	0.00
Iron	0–5	3.21	0.61	>100% recommendations	5	11.9
				<100–50% recommendations	3	89.00
				<50% recommendations	1	0.00
				0% recommendations	0	0.0
Calcium	0–5	3.52	1.33	>100% recommendations	5	39.00
				<100–50% recommendations	3	48.80
				<50% recommendations	1	12.20
				0% recommendations	0	0.00
Vitamin C	0–5	4.44	1.07	>100% recommendations	5	76.50
				<100–50% recommendations	3	19.10
				<50% recommendations	1	4.30
				0% recommendations	0	0.00
Moderation	0–30	17.52	6.25			
Total fat	0–6	1.96	2.36	<20% of total energy/day	6	19.50
				>20–30% of total energy/day	3	26.20
				>30% of total energy/day	0	54.30
Saturated fat	0–6	1.68	2.33	<7% of total energy/day	6	17.70
				>7–10% of total energy/day	3	20.70
				>10% of total energy/day	0	61.60
Cholesterol	0–6	4.66	2.13	<300 mg/day	6	68.30
				>300–400 mg/day	3	18.90
				>400 mg/day	0	12.80
Sodium	0–6	5.18	1.57	<2400 mg/day	6	76.20
				>2400–3400 mg/day	3	20.10
				>3400 mg/day	0	3.70
Empty calorie food	0–6	4.04	2.32	<3% of total energy/day	6	54.10
				>3–10% of total energy/day	3	27.90
				>10% of total energy/day	0	18.00
Overall balance						
Macronutrient ratio (carbohydrate:protein:fat)	0–10	2.66	2.26	55–65:10–15:15–25	6	4.90
				52–68:9–16:13–27	4	7.30
				50–70:8–17:12–30	2	56.70
				Otherwise	0	31.10
Fatty acid ratio (PUFA MUFA/SFA)	0–4			PUFA/SFA = 1–1.5 and MUFA/SFA = 1–1.5	4	21.30
				PUFA/SFA = 0.8–1.7 and MUFA/SFA = 0.8 = 1.7	2	1.20
				Otherwise	0	77.50

**Table 4 healthcare-11-00138-t004:** Association between Diet Quality Index (DQI-I) and socio-demographic variables.

		N	%	DQI	SD	*p* *
BMI (kg/m^2^)	<24.99	236.00	59.10	54.61	11.67	0.571
	25.00–29.99	103.00	26.60	53.15	11.53	
	>30.00	61.00	15.30	52.16	10.46	
According to weight	Underweight	52.00	13.00	54.00	10.04	0.781
	Normal	232.00	58.00	55.41	11.42	
	Overweight/obesity	116.00	29.00	54.58	8.31	
Physical activity level	Sedentary	331.00	82.90	54.15	10.97	0.521
	Active	69.00	17.10	52.59	13.79	
Education level of father	Low	143.00	35.70	54.39	10.75	0.555
	Medium	108.00	26.90	55.37	11.16	
	High	84.00	21.10	52.00	12.76	
	Do not know	65.00	16.20	53.16	11.10	
Educational level of mother	Low	68.00	16.90	56.96	11.68	0.411
	Medium	167.00	41.90	52.71	11.63	
	High	86.00	21.40	54.90	11.60	
	Do not know	79.00	19.80	53.20	10.50	
Location lunch	Home	136.00	34.00	55.65	10.86	0.154
	School	264.00	66.00	52.94	11.65	
Breakfast considered important	Yes	180.00	45.00	54.23	10.63	0.711
	No	220.00	55.00	53.56	12.10	
Knowledge in nutrition	Excellent	44.00	11.00	52.75	4.65	0.305
	Well	62.00	15.50	51.69	10.03	
	Medium	250.00	62.50	53.71	10.72	
	Low	44.00	11.00	49.33	12.99	

* Test T/ANOVA. T test, *p* < 0.001.

**Table 5 healthcare-11-00138-t005:** DQI-I scores and selected food and nutrient intakes by Diet Quality Index-International (DQI-I) score category in Mexico.

	Diet Quality Index-International (DQI-I)		
	≤49.00	50.00–59.00	60.00+
N	138	130	132
**DQI score**			
DQI-I. total	41.81	53.81	66.89
Variety score	3.42	9.37	14.10
Adequacy score	21.65	24.97	28.45
Moderation score	14.95	16.79	20.83
Overall balance score	1.79	2.69	3.51
**Food and nutrient intake**			
Fruit (servings/day)	2.19	5.03	7.59
Vegetable (servings/day)	1.19	2.50	5.46
Dietary fiber (g/day)	14.28	16.62	17.25
Energy from fat (%)	342.11	329.53	287.90
Energy from SFA (%)	132.31	116.15	78.74
Riboflavin (mg/day)	2.81	2.96	4.48
Vitamin C (mg/day)	111.29	118.04	112.46
Vit. A (ug/day)	798.84	753.65	1659.65
Vit. D (ug/day)	3.85	3.57	4.30
Vit. E (mg/day)	10.94	13.95	18.22
Calcium (mg/day)	741.78	770.60	695.99
Iron (mg/day)	9.92	12.62	14.59
Sodium (mg/day)	3503.95	3809.11	5283.35
Zinc (mg/day)	7.06	9.18	14.55
Se (ug/day)	52.32	52.11	48.30
Folate (ug/day)	223.45	244.39	229.60
Father studies			
Primary (%)	28.10	34.60	32.10
Media (%)	22.80	28.80	32.10
University (%)	29.80	25.00	22.60
NS/NC (%)	19.30	11.50	13.20
Mother studies			
Primary (%)	15.80	11.50	17.00
Media (%)	47.40	50.00	43.40
University (%)	19.30	17.30	22.60
NS/NC (%)	17.50	21.20	17.00

## Data Availability

The data presented in this study are available on request from the corresponding author M.M.-A. (mariscal@ugr.es).

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
