# Peer review of "Estimation of the Quality of the Diet of Mexican University Students Using DQI-I"

_healthcare, 2023, doi:10.3390/healthcare11010138_

Round 1

Author Response

RESPONSE:

Line 11-33: corrected.

Line 36: corrected.

Line 56: corrected.

Line 66: corrected.

Line 89: corrected.

Line 94: corrected.

Line 201: corrected.

Line 205: corrected.

Line 217: corrected.

Line 354: corrected.

Line 382: corrected.

Line 398: corrected.

Reviewer 2 Report

This study investigated the impact of a westernized diet on the Mexican population in, particularly young adult women (n=400) whose health indicates were assessed utilizing diet Qualities indices-international (DQI-I).

They provided detailed information on diet assessment and the interpretation was correct in the scientific report, including discussion align with the significance of their findings compared to cited references along with presenting their big data and analytic tool with statistical methodology. It is well-written and demonstrated clearly. However, there is room for improvement by correcting the model with gender and compared to the traditional lifestyle of Mexican Food and debating some issues like behavioral traits in the future.

Very impressive Job! Well Done.

Minor typo:

I am afraid Reference No 28 was missing.

Author Response

RESPONSE: Thank you very much for the comments on the review. Reference 28 has been included in line 152.

Reviewer 3 Report

Authors have written an interesting article about important topic. However, there are few things to be considered.

Major concerns:

DQI-I seems to be out of date when it gives points for eating meat (overall, poultry), egg and dairy products (both in the overall variety score and the score for variety of protein sources). It is widely known that eating meat is not healthy. WHO has classified red meat as potentially carcinogenic and all processed meat as carcinogenic. Many studies (for example Epic-Oxford and Adventist Health Studies) have also shown that eating meat (and also other animal products) increases mortality.

See for example:

https://www.researchgate.net/publication/328030518_Health_and_sustainability_outcomes_of_vegetarian_dietary_patterns_a_revisit_of_the_EPIC-Oxford_and_the_Adventist_Health_Study-2_cohorts

https://bmcmedicine.biomedcentral.com/articles/10.1186/s12916-021-01922-9

https://www.thelancet.com/journals/lanonc/article/PIIS1470-2045(15)00444-1/fulltext

https://www.annalsofoncology.org/article/S0923-7534(19)32133-7/fulltext

https://pubmed.ncbi.nlm.nih.gov/35790788/

https://www.ahajournals.org/doi/10.1161/JAHA.120.017066

It thus seems that DQI-I should be either abandoned or at least modified so that it doesn´t give points from animal products (except low or moderate amount of fish and non-fat dairy). Please clarify and discuss this extensively in introduction, discussion and conclusions. It is crucial because otherwise we woukd be using and recommending a tool (DQI-I) which encourages people to use meat and other unhealthy animal products (which would be harmful not only for their health but also for environment).

Lines 230-238: See above. Humans get all the essential amino acids also by combining different plant sources of protein (for example 1. grains and/or 2. legumes and/or 3. seeds or nuts). So it is misleading to call animal proteins "complete". It gives a picture that plant proteins are missing something which is not true if you combine them. Please correct.

Lines 300-304: It essential not to recommend nutritional education to increase variety in animal based protein sources. It should be made clear that it doesn´t bring any health benefits. But increasing variety of plant based protein sources is definately good thing to recommend. Please correct.

Minor concerns:

Line 80. Sentence begins with small ”a” and verb is missing.

Line 88: Please remove phrase "extensively used".

Line 97: Third questionnaire was a food diary or what? Line 101: There is extra dot after ”(model SECA 214; 20-207 cm)”. Should be comma? Line 103: Dot after ”normal weight” should also be comma. Line 155: Mean intakes are compared to Dietary Guidelines for Americans, not to international recommendations. Please correct. Line 173: ”o” should be ”of” Line 198-199: ”however, the average studies of the 198 mother group the highest follow-up by the university population of the DQI-I (Table 5).” - What does this mean? Please clarify. Table 1. Word ”Carbohydrate” with small letter.

Table 3. "<50100% recommendations" should be "50100% recommendations"?

Table 3. Fatty acid ratio has criteria written two times (in the table and under the table), is this necessary?

Tables 3 and 4. Please separate different sections with horizontal lines. Table 5. ”Estudios madre” should be ”Mother’s education”?

Author Response

REVIEWER 3

Authors have written an interesting article about important topic. However, there are few things to be considered.

Major concerns:

DQI-I seems to be out of date when it gives points for eating meat (overall, poultry), egg and dairy products (both in the overall variety score and the score for variety of protein sources). It is widely known that eating meat is not healthy. WHO has classified red meat as potentially carcinogenic and all processed meat as carcinogenic. Many studies (for example Epic-Oxford and Adventist Health Studies) have also shown that eating meat (and also other animal products) increases mortality.

See for example:

https://www.researchgate.net/publication/328030518_Health_and_sustainability_outcomes_of_vegetarian_dietary_patterns_a_revisit_of_the_EPIC-Oxford_and_the_Adventist_Health_Study-2_cohorts

https://bmcmedicine.biomedcentral.com/articles/10.1186/s12916-021-01922-9

https://www.thelancet.com/journals/lanonc/article/PIIS1470-2045(15)00444-1/fulltext

https://www.annalsofoncology.org/article/S0923-7534(19)32133-7/fulltext

https://pubmed.ncbi.nlm.nih.gov/35790788/

https://www.ahajournals.org/doi/10.1161/JAHA.120.017066

It thus seems that DQI-I should be either abandoned or at least modified so that it doesn´t give points from animal products (except low or moderate amount of fish and non-fat dairy). Please clarify and discuss this extensively in introduction, discussion and conclusions. It is crucial because otherwise we woukd be using and recommending a tool (DQI-I) which encourages people to use meat and other unhealthy animal products (which would be harmful not only for their health but also for environment).

RESPONSE: We agree with your comment about the high consumption of red meat and low consumption of proteins of vegetable origin, as referred to by different researchers cited by you. The DQI-I, (Kim et al, 2003) was proposed based on the Western Diet and compared the US diet with the Chinese diet. Our group already introduced modifications to this index when it was applied to the Mediterranean population.

Mariscal-Arcas M, Velasco J, Monteagudo C, Caballero-Plasencia MA, Lorenzo-Tovar ML, Olea-Serrano F. Comparison of methods to evaluate the quality of the Mediterranean diet in a large representative sample of young people in Southern Spain. Nutr Hosp. 2010 Nov-Dec;25(6):1006-13.

Mariscal-Arcas M, Romaguera D, Rivas A, Feriche B, Pons A, Tur JA, Olea-Serrano F. Diet quality of young people in southern Spain evaluated by a Mediterranean adaptation of the Diet Quality Index-International (DQI-I). Br J Nutr. 2007 Dec; 98(6):1267-73. doi: 10.1017/S0007114507781424.

But in this paper, the DQI published by KIM et al. has been taken as a reference, since it is a widely used index and allows us to establish a measure of the diet quality of a population from northern Mexico (State of Chiuahua) and in the papers published so far describe it as being close to the Western Diet, and about which not much information has been published.

In our papers we do not intend to promote a certain dietary pattern, only to classify the population under study in a reference range. In fact, this population does not follow exhaustively the reference diet of Kim et al. The mean value of the DQI is 53.86 (SD: 11.43). The Variety parameter is low, 26% of the study population only includes one protein source, this has already been discussed in the paper and it may mean that this population does not follow the optimal parameters of the applied index. The index and we do not advocate the consumption of meat and derivatives, simply a review of some results is made. The paper presented is limited to describing under an international index that the study population is monitored and being able to place it in an international ranking with a reference pattern. The results obtained show approximately 50% follow-up of the applied index and we do not recommend following or not this dietary mode.

Lines 230-238: See above. Humans get all the essential amino acids also by combining different plant sources of protein (for example 1. grains and/or 2. legumes and/or 3. seeds or nuts). So it is misleading to call animal proteins "complete". It gives a picture that plant proteins are missing something which is not true if you combine them. Please correct.

RESPONSE: It can be defined: 1) A quality protein will be one that provides all the nutrients that we should expect from said food 2) Complete Protein. It refers to the protein in whose amino acid profile those elements that our body is not capable of manufacturing are available: essential amino acids.

However, to avoid possible confusion, Proteins from meat and other animal products are complete proteins. This means they supply all of the amino acids the body can't make on its own. has been removed.

Lines 300-304: It essential not to recommend nutritional education to increase variety in animal based protein sources. It should be made clear that it doesn´t bring any health benefits. But increasing variety of plant based protein sources is definately good thing to recommend. Please correct.

RESPONSE: Has been added: “(for example, introduce into your diet combinations of grains and legumes, grains and grains seeds and nuts, legumes seeds and nuts)”.

Minor concerns:

Line 80. Sentence begins with small ”a” and verb is missing.

RESPONSE: corrected the questionnaire was applied……..

Line 88: Please remove phrase "extensively used".

RESPONSE: Removed.

Line 97: Third questionnaire was a food diary or what?

RESPONSE: The third questionnaire was a three-day 24HR for three non-consecutive days, including one non-working day. It was administered once a month between February and May. At the end, similar information was obtained for three consecutive months.

Line 101: There is extra dot after” (model SECA 214; 20-207 cm)”. Should be comma?

RESPONSE: corrected.

Line 103: Dot after ”normal weight” should also be comma.

RESPONSE: corrected.

Line 155: Mean intakes are compared to Dietary Guidelines for Americans, not to international recommendations. Please correct.

RESPONSE: corrected.

Line 173: ”o” should be ”of”

RESPONSE: corrected.

Line 198-199: ”however, the average studies of the 198 mother group the highest follow-up by the university population of the DQI-I (Table 5).” - What does this mean? Please clarify.

RESPONSE: there is an error in the transcription of the text should say; however, mothers with a medium level of education are the ones who mostly follow DQI-I (Table 5).

Table 1. Word ”Carbohydrate” with small letter.

RESPONSE: corrected.

Table 3. "<50–100% recommendations" should be "50–100% recommendations"?

RESPONSE: corrected <100-50%

Table 3. Fatty acid ratio has criteria written two times (in the table and under the table), is this necessary?

RESPONSE: removed.

Tables 3 and 4. Please separate different sections with horizontal lines.

RESPONSE: done.

Table 5. ”Estudios madre” should be ”Mother’s education”?

RESPONSE: Following the section referring to the father, is mother studies; Corrected.

Round 2

Reviewer 3 Report

Authors have improved their article. However, there are still some important things to be corrected.

-------------------------------------------------------------------------

1. Most important thing is to write out  that DQI-I is out of date when it gives points for eating meat, poultry, egg yolks and fatty dairy products (both in the overall variety score and the score for variety of protein sources). And not only because of the culture of the population studied, but because there is plenty of evidence that these foods are not healthy.

For example: "DQI-I gives points for eating meat, poultry, egg yolks and fatty dairy products. It is widely known that eating meat is not healthy. WHO has classified red meat as potentially carcinogenic and all processed meat as carcinogenic. Many studies [add references for example to Epic-Oxford and Adventist Health Studies, UK Biopank study] have also shown that eating meat (and also other animal products) increases mortality. According to this knowledge of associations between diet and health, it can´t be stated that DQI-I is a good way measure the quality of diet."

2. Lines 307-309 is better now. However, it is a bit hard to read / underestand. Maybe it would be better to state something like; "(for example, combine legumes, soy products, grains, nuts and seeds in your diet)".

-------------------------------------------------------------------------

Minor concerns: Line 14: Dot is missing after ”girls”

Author Response

REVIEWER 3

Authors have improved their article. However, there are still some important things to be corrected.

-------------------------------------------------------------------------

  1. Most important thing is to write out that DQI-I is out of date when it gives points for eating meat, poultry, egg yolks and fatty dairy products (both in the overall variety score and the score for variety of protein sources). And not only because of the culture of the population studied, but because there is plenty of evidence that these foods are not healthy.

For example: "DQI-I gives points for eating meat, poultry, egg yolks and fatty dairy products. It is widely known that eating meat is not healthy. WHO has classified red meat as potentially carcinogenic and all processed meat as carcinogenic. Many studies [add references for example to Epic-Oxford and Adventist Health Studies, UK Biopank study] have also shown that eating meat (and also other animal products) increases mortality. According to this knowledge of associations between diet and health, it can´t be stated that DQI-I is a good way measure the quality of diet."

RESPONSE: The authors would like to thank Reviewer 3. We have included the text proposed by the Reviewer at the end of the discussion, adding the requested references both in the text and in the bibliography section. We think that the added text improves the final discussion of the manuscript.

  1. Lines 307-309 is better now. However, it is a bit hard to read / underestand. Maybe it would be better to state something like; "(for example, combine legumes, soy products, grains, nuts and seeds in your diet)".

RESPONSE: The authors have revised the added text, thinking that the Reviewer is actually referring to the new lines 522-524 of the manuscript. In that case, we agree with the reviewer that it is better now.

-------------------------------------------------------------------------

Minor concerns: Line 14: Dot is missing after ”girls”

RESPONSE: A dot has been inserted after "girls".
